

# A new drought index fitted to clay shrinkage induced subsidence over France: benefits of interactive leaf area index

Sophie Barthélemy[1,2,3], Bertrand Bonan[1], Jean-Christophe Calvet[1], Gilles Grandjean[2], David Moncoulon[3], Dorothée Kapsambelis[3], and Séverine Bernardie[2]

[1]CNRM, Université de Toulouse, Météo-France, CNRS, Toulouse, France

[2]Bureau de Recherches Géologiques et Minières (BRGM), Orléans, France

[3]Caisse Centrale de Réassurance (CCR), Dpt R&D Modeling Cat & Agriculture, Paris, France

*Correspondence to*: Jean-Christophe Calvet (jean-christophe.calvet@meteo.fr)

**Abstract.** Clay shrinkage, which consists of a reduction in the volume of clay soils during dry periods, can affect buildings and cause subsidence damage. In France, losses due to subsidence are estimated at more than 16 billion € for the period 1989-2021 (CCR, 2021), and are expected to increase under the effect of climate warming. This work aims to improve the current understanding of the conditions triggering subsidence by proposing an innovative drought index. We use a daily Soil Wetness Index (SWI) to develop a new annual drought index that can be related to subsidence damage. The SWI is derived from simulations of soil moisture profiles from the Interactions between Soil, Biosphere, Atmosphere (ISBA) land surface model developed by Météo-France. The ability of the drought index to correlate with insurance claims data is assessed by calculating the Kendall rank correlation over twenty municipalities in France. The insurance data, aggregated by year and municipality, are provided by the Caisse Centrale de Réassurance (CCR). A total of 1200 configurations of the drought index are considered. They are generated by combining different calculation methods, ISBA simulation settings, soil model layers, and drought percentile thresholds. The analysis includes a comparison with the independent claim data of six additional municipalities, and to a record of official "CatNat" decrees, useful for the analysis. The best results are obtained for drought magnitudes based on SWI values of the 0.8 m to 1.0 m deep soil layer, an ISBA simulation with interactive leaf area index (LAI), and consideration of low drought SWI percentile thresholds. Comparison with claim data shows that drought magnitude is able to identify subsidence events while being spatially consistent. This drought magnitude index provides more insight into subsidence triggers while benefiting from advanced land surface modeling schemes (interactive LAI, multi-layer soil). This work paves the way for more reliable damage estimates.

## 1 Introduction

Clay shrink-swell is the change in volume of clayey soils in response to changes in soil moisture. This phenomenon is related to the high affinity of certain clay minerals commonly found in soils for water molecules.



These minerals, structured in aggregates, swell and shrink under wet and dry conditions, respectively (Bronswijk, 1989). In temperate European regions such as in France, soils are in a hydrated state for most of the year, which means that exceptional movements occur during droughts (Cojean, 2007). On the other hand, surface soil moisture variations result from soil-atmosphere interactions. Precipitation brings in water, only some of which infiltrates into the soil, while evaporation causes water loss. Vegetation affects both water inputs by intercepting precipitation and preventing some of it from reaching the soil surface, and water outputs through root water uptake and transpiration (Tessier et al., 2007).

Shrinking ground movements caused by droughts can affect buildings through their foundations, causing what is known as subsidence damage (Doornkamp, 1993). The damage is physically caused by the inability of a building to accommodate a relative displacement of its various parts, leading to structural failure. This relative displacement observed during droughts is caused by the different behavior of exposed shrinking and sheltered soils. The latter are immobile due to their constant moisture content. Local factors that increase spatial differences in soil moisture, such as the presence of vegetation,significant exposure to solar radiation, or a deficient domestic drainage system are known to exacerbate subsidence (Page, 1998).

In France, subsidence losses were reported after the 1976 summer drought. A few years later, hundreds of thousands of houses were affected by the 1989 drought, which led to the inclusion of this peril in the French National Natural Disaster Compensation Scheme (CatNat) (Salagnac, 2007). Today, subsidence is still covered under the CatNat regime, where it is the second most costly peril after floods. Between 1989 and 2021, losses amounted to more than 16 billion € (CCR, 2021). Historically, the southwest of France has been the most exposed to subsidence. However, a significant number of events have recently been reported in the Northeast since 2016 (CCR, 2022), showing a geographical expansion of the importance of subsidence in France. Since then, the cost of subsidence costs has averaged €1 billion per year, and even exceeded 2 billion € in 2022, which was marked by an exceptional drought (Toreti et al., 2022).

Several studies on subsidence damage (e.g. Crilly, 2001) have shown that single-family houses are the most vulnerable type of building because they tend to have shallow foundations. In 2019, approximately 19 million single-family houses have been identified in France, more than half of which are located in zones with a medium to high exposure to clay shrinkage (MTES, 2021). In addition, research led by Météo-France, the French meteorological service, within the CLIMSEC project (https://www.umr-cnrm.fr/spip.php?article605) on the impact of climate change on soil water resources provided insight into climate change-induced drought trends for the country (Vidal et al., 2010, 2012). It was shown that the duration and spatial extent of drought events affecting soils are bound to increase as a result of rising air temperatures and changes in the precipitation regime.

In this context, both policy makers and the insurance industry need more accurate predictions of subsidence losses. Research is needed to develop a better understanding of this phenomenon and of its financial consequences.

Subsidence depends on predisposing factors, such as the type of clayey soil or land use, and on triggering factors. The first studies to investigate the dependence of clay shrinkage-induced subsidence on climate (Harrison et al., 2012; Corti et al., 2009, 2011) confirmed the ability of drought to trigger subsidence. These analyses were based on precipitation and air temperature data and recommended the use of land surface models (LSMs) as they provide more detailed information on droughts. In the



wake of these findings, several works focusing on subsidence in France have used the soil wetness index (SWI) outputs of the ISBA LSM (Noilhan and Planton, 1989; Noilhan and Mahfouf, 1996) developed by Météo-France, in a simplified
configuration (monthly averages, homogeneous vegetation and soil). This product called "SWI uniforme", is also used for drought monitoring in the CatNat regime. However, the criterion based on this "SWI uniformé" for the promulgation of decrees has changed several times, making it impossible to link the history of decrees to quantitative drought characteristics. Using this "SWI uniforme", a number of studies have assessed the impact of climate change on subsidence costs (Gourdier and Plat, 2018; CCR and Météo-France, 2018; Covéa and RiskWeatherTech, 2022). All of these papers concluded that subsidence-
related costs would increase, but no consensus was reached on the magnitude of the increase. Other studies attempted to improve the predictive power of subsidence damage models using advanced machine learning techniques (Heranval et al., 2022; Ecoto and Chambaz, 2022; Charpentier et al., 2022).

Motivated by the above context, a new drought index is proposed in this study, which is adapted to clay shrinkage-induced subsidence damage. The novelty of the present work lies in the analysis of simulations using a version of the ISBA LSM with
an extended vegetation representation. The index is based on soil moisture simulations from a version of ISBA capable of simulating the details of the soil moisture profile together with an interactive vegetation leaf area index (LAI), and is adjusted by comparison with insurance data. The objective is to determine (1) which soil layers should be considered for subsidence damage prediction, and (2) the extent to which interactive LAI is needed.

In section 2, all data sets and the methodology for computing the drought index are presented. In section 3, we detail the
obtained optimal drought index and its relationship to the subsidence damage data. Comparison with an independent dataset and with a dataset of CatNat decrees is included. All of these results are discussed in section 4 before concluding in section 5.

## 2 Data and methods

The objective of this work is to create a new soil drought index that is adapted to induced subsidence losses. We here describe how the indices are computed from "LAI_clim" and "LAI_model" soil moisture simulations, and how they are evaluated
through a pairwise ranking with the insurance damage data. A comparison with independent data is included in the analysis. The methodology is summarized as a flowchart in Fig. 1.

### 2.1 Soil moisture simulations

Soil moisture is simulated by ISBA within version 8.1 of the SURFEX (SURFace Externalisée) modeling platform developed by Météo-France for numerical weather prediction and climate modeling (Masson et al., 2013; Le Moigne et al., 2020). ISBA
calculates surface water and energy budgets in response to an atmospheric forcing. Soil moisture at a given time results from the balance between water inflows from precipitation, and outflows, through runoff, infiltration, and evaporation. ISBA covers the entire French metropolitan area, on an 8 km meshed grid. It is used operationally for monitoring water resources in the SAFRAN-ISBA-MODCOU (SIM) hydrological suite (Habets et al., 2008), together with a meteorological analysis and a



hydrogeological model. ISBA is used here "offline", i.e. there is no feedback from the surface to the atmosphere. The SAFRAN
reanalysis (Durand et al., 1993; Quintana-Seguí et al., 2008) is used as the atmospheric forcing.

A multilayer version of ISBA, ISBA-DIF (Boone et al., 2000; Decharme et al., 2011), with a diffusive scheme is used. The
approach is to divide the soil column into layers whose respective thicknesses increase with depth to better represent
hydrological processes, since important water and temperature gradients at the surface require a finer mesh. The number of
layers considered is proportional to the rooting depth of the vegetation, reaching a maximum of 10 layers (adding up to 2 m
depth) in the case of trees. It is important to analyze layers at different depths because moisture variations depend on both the
distance to the surface and the vegetation roots.

Soil hydrologic properties, such as water holding capacity, change with soil texture. In ISBA, texture is considered to be
homogeneous throughout the whole soil column for each grid point. This texture is represented by clay, sand, and silt contents
derived at the kilometer resolution from the Harmonized World Soil Database (HWSD) version 1.2 (Nachtergaele et al., 2012).
To account for spatial variability within a grid cell, ISBA first runs separately for 12 generic plant functional types called
patches (bare soil, rock, permanent snow, deciduous broadleaf trees, evergreen broadleaf trees, coniferous trees, $C_3$ crops, $C_4$
crops, irrigated $C_4$ crops, grassland, tropical grassland, and wetlands). The runs are then aggregated by averaging the output
variables weighting each patch by its respective fraction in the grid cell. The maximum soil depth of a patch depends on the
rooting depth of the corresponding vegetation type and varies from patch to patch. The geographic distribution of the patches
as well as the land surface parameters are obtained from the ECOCLIMAP-II database at kilometer resolution (Faroux et al.,
2013). The present work focuses on layers simulated for all patches, as we aim to assess differences induced only by depth.
These conditions were met for four soil layers: 0.2-0.4 m, 0.4-0.6 m, 0.6-0.8 m, 0.8-1.0 m, soil layers 5, 6, 7, and 8, respectively.
Accordingly, we based our analysis on these four model layers.

Because vegetation plays an important role in near-surface soil moisture variations, the ISBA model includes a representation
of photosynthesis using a $CO_2$-responsive stomatal conductance scheme, ISBA-A-gs (Calvet et al., 1998). This scheme has
been improved over time to account for specific plant responses to drought (Calvet, 2000; Calvet et al., 2004).

In the operational SIM product (SURFEX v8.0), the leaf area index (LAI), which quantifies the leaf surface available for
exchange with the atmosphere, is prescribed from ancillary data. For each grid point, the LAI changes every 10 days, and the
cycle is repeated every year, based on a climatology established from SPOT/VGT satellite data for the period  from 1999 to
2005. A first simulation, with this configuration is tested in this study and is called LAI_clim.

In the ISBA-A-gs configuration the LAI is not prescribed, but simulated from the modeled leaf biomass, taking into account
the mass-based leaf nitrogen concentration (Calvet and Soussana, 2001). Phenology is driven by photosynthesis. Since the
modeled photosynthesis depends on soil moisture, leaf temperature, solar radiation, and air humidity, all environmental
conditions can affect the simulated LAI. Based on this concept, the LAI is updated daily. The modeled LAI has been validated
at global (Gibelin et al., 2006), continental (Szczypta et al., 2014), and regional (Brut et al., 2009) scales. In this work, a second
simulation is tested with this configuration, which we call LAI_model.



Each simulation provides hourly volumetric soil moisture data for four 0.2 m thick soil layers from 0.2 to 1.0 m depth. Since we are focusing on more long-term droughts, the diurnal periodicity of the data is not required, and the hourly soil moisture values are averaged to daily values. In order to consider a single definition of drought, independent of the fact that water

holding capacity varies with soil texture, we convert volumetric soil moisture to soil wetness indices (SWI) by rescaling soil moisture between wilting point $w_{\text{wilt}}$ and field capacity $w_{\text{fc}}$. These two quantities are derived from the model soil texture through soil pedotransfer functions (Boone et al., 1999; Decharme et al., 2013). We refer to the SWI of layers 5, 6, 7 and 8 as SWI5, SWI6, SWI7 and SWI8, respectively.

**2.2 Normalized insurance data**

The insurance data used in this study are provided by the Caisse Centrale de Réassurance (CCR), the French public reinsurance company that manages a large database of its clients' contributions. It consists of the number of subsidence claims aggregated per year and per municipality, from 2000 to 2018. By "municipality", we mean a commune, the smallest administrative unit in France. The representativeness of the data varies over the years and the municipalities considered, depending on the proportion sent by insurers. Total and average annual costs were also provided by CCR, butt we focused on the number of

claims, which gave more satisfactory results. The data is aggregated at the municipal level for confidentiality reasons. In addition, this is an official and frequently used zoning classification, so there is a lot of additional data available.

For normalization purposes, the large number of subsidence claims is divided by the number of single-family houses in the municipality located in areas prone to subsidence. This information comes from a dataset developed by the French Ministry of Ecology (MTES/CGDD/SDES) (Nouveau zonage d'exposition au retrait-gonflement des argiles : plus de 10,4 millions de

maisons individuelles potentiellement très exposées, 2023). It provides the number of houses per municipality in different exposure classes (no exposure, low and medium to high). The zoning is based on a map of exposure to clay shrink-swell developed by the French Geological Survey (BRGM) (BRGM, 2023), coupling a national susceptibility map (based on geotechnical data and 1:50000 geological maps) and insurance statistics. We call the new variable obtained after normalization the "normalized number of claims".

In this study, we use data from 26 municipalities, the location of which is shown in Fig. 2. Although the spatial resolution of the insurance data leads us to an analysis at the municipal level, we generalize the results by repeating the process for areas exposed to different climates. The study is limited to a selection of municipalities with a significant history of clay shrink-swell losses, since only the dependence on climate is studied (see Section 3).

**2.3 Natural disaster (CatNat) decree database**

In France, financial losses to insured property following a natural catastrophe may be covered by a national compensation scheme, the "Régime d'indemnisation des Catastrophes Naturelles" (CatNat regime). Compensation under this scheme is, with a few exceptions, subject to the official recognition of the state of natural disaster through the publication of a decree, an "arrêté de reconnaissance de l'état de catastrophe naturelle" (CatNat decree), for a given year and municipality. The GASPAR





database provided by the French government (Georisques, 2023) lists all the CatNat decrees promulgated since the creation of the system in 1982. This information is relevant for our research because subsidence damages are only compensated by this scheme since 1989.

After 2000, the publication of a CatNat decree is conditioned by a geotechnical criterion and by a meteorological criterion. The geotechnical condition requires the municipality to have at least 3 % of its surface mapped as shrinkable clay, based on the BRGM exposure to clay shrink-swell map. The meteorological condition aims to identify exceptional periods of drought. Its current version requires the 3-month average of the "SWI uniforme" to be the lowest or second lowest of a 50-year reference associated with the same month. This criterion has evolved several times since the creation of the system as the phenomenon has been studied. The condition established in 2000 is a "winter" criterion, based on the rainfall of the winter preceding the drought event. Later, the timing of the successive drought events of 2003, 2011 and 2018 led to the creation of "summer", "spring" and "fall" criteria, that were added to the existing.

## 2.4 Drought indices

Providing a universal and quantitative definition of drought is challenging because drought can affect different levels of the hydrological cycle: precipitation, soil moisture and streamflow (Dracup et al., 1980). Numerous indices characterizing different types of drought are available in the literature. For example, the Standardized Precipitation Index (SPI) (McKee et al., 1993), quantifies meteorological droughts by fitting accumulated precipitation at different time scales to a given distribution. The same methodology has already been adapted for agricultural droughts, where calculations are based on the SWI, creating a Standardized Soil Wetness Index (SSWI) (Vidal et al., 2010). As Charpentier et al. (2022) recall from the conclusions of Soubeyroux et al. (2012), the SPI and the SSWI provide complementary information: for example, the 2003 drought over France, which is more of a heatwave than a lack of precipitation, is only detected by the SSWI.

Given the nature of the clay shrink-swell phenomenon, we focus here on droughts affecting soil moisture by analyzing the SWI outputs of the ISBA model. LSMs are capable of accounting for the atmospheric forcing, soil and vegetation characteristics and initial conditions. They are the most reliable tools for studying soil moisture variations.

Subsidence damage affects several regions located throughout France, and is therefore subject to different climates (see Fig. 2 for a climatic zoning of the country). We consider the tipping point for subsidence not as an absolute value of soil moisture, but rather as a measure of the deviation from the mean. This definition is consistent with the physical mechanism of subsidence damage, which is related to a difference in soil moisture content relative to its average value (Section 1).

In this study, we define droughts as periods in which the value of the daily SWI is below a threshold value. The latter corresponds to a certain percentile of the empirical SWI distribution calculated over a given ISBA grid point. Based on *in situ* observations in the United States and Canada, Ford et al. (2016) showed that the record length required to obtain stable daily soil moisture distributions ranges from 3 to 15 years. Assuming that these results are applicable to France, our study period of 19 years is sufficient to obtain reliable SWI percentiles. The use of percentiles also avoids the need to standardize SWI values, as the shape of their distributions is quite variable depending on the local climate (D'Odorico et al., 2000; Vidal et al., 2010).



In this study we deliberately make no assumption about the triggering drought frequency, and therefore consider thresholds ranging from the first percentile to the median.

Once drought periods have been identified using the daily SWI percentile criterion, several options are considered to quantify

the drought events. For this purpose, the characteristics of drought events are described in Vidal et al. (2010): severity, duration and magnitude. We apply this concept to our definition of drought:

- Severity is the maximum deviation of the SWI from the threshold,
- Duration is the number of days below the threshold,
- Magnitude is the sum of the daily SWI deficit values (equivalent to an integral) below the threshold.

All three indices are considered. Given the temporal resolution of the insurance data, we assume only one major drought event per year, as we are working on an annual scale. Therefore, a single index value is calculated each year.

For each municipality, the three indices are computed for two model configurations (LAI_clim and LAI_model), four model soil layers 5 to 8, and 50 SWI percentile values (from 1 to 50). This corresponds to 1200 simulations per municipality.

## 2.5 Pair-wise ranking of years

To assess the ability of the drought indices to represent subsidence damage for a given year, we compute the Kendall rank correlation coefficient (Wilks, 2006) between the two variables, over the entire study period. The Kendall rank correlation test compares the respective ranks of paired data by counting concordant and discordant pairs. It produces a coefficient ranging from 0 to 1, the Kendall tau, and a p-value (the null hypothesis being that the variables are not correlated). This test has the advantage of being nonparametric and robust to extreme values.

To maximize the number of observations used to compute the Kendall tau, we merge data from multiple municipalities to form regional subsets. An association is performed to pair the municipal insurance data with the soil moisture available on the ISBA grid. We compare the soil and vegetation characteristics of all the ISBA cells that are less than 2.5 km from the municipality boundary, ultimately selecting the cell with the highest grass and clay content (most representative of a damaged building environment) as the most representative cell. We choose to base the selection on soil and vegetation rather than distance

because we believe that outside of mountainous areas, the atmospheric forcing does not change drastically for two neighboring cells.

For each regional subset, the Kendall tau is computed for all the 1200 possible drought index configurations. The highest Kendall tau values are used to identify optimal configurations. Correlation calculations are performed sequantially for regional subsets and are combined by calculating a mean Kendall tau.

## 2.6 Validation

The best drought index configuration is then applied to independent data to verify that we avoid over-fitting. For this purpose, the drought index is computed for all the municipalities not used in the calibration (forming a validation set) and confronted with the corresponding normalized number of claims. For further analysis, the optimal drought indices of both calibration and





validation sets are sorted into classes and the distributions of the normalized damage counts are compared. Four drought index
classes are defined from the calibration set: a first class that includes all zero values (no drought), and three other classes that
divide the rest of the population into equal groups. The class delineation is applied to the validation set.

Finally, both the drought index and the normalized insurance data are compared to the set of historical CatNat decrees.

For each subset, we assess the ability of the optimal drought index to predict claims by counting contingencies and deriving
scores (Wilks, 2006). First, we transform quantitative optimal drought magnitudes and normalized claim counts into Boolean
occurrences of predictions and observations, respectively. The correspondences for each subset are reported in a contingency
table. We refer to the four metrics in the contingency table as true positives, true negatives, false positives and false negatives
(TP, TN, FP, and FN, respectively). The performance of the index in identifying damage patterns is then evaluated for each set
by calculating scores. From the variety of scores available in the literature, we selected the Proportion Correct (PC), Bias (B),
False Alarm Ratio (FAR), Probability Of Detection (POD) and Probability Of False Detection (POFD). Each of these metrics
is described in detail in Table 1.

## 2.7 Study area

The calibration set consists of five subsets of four municipalities distributed throughout France, in different climatic settings
(continental, modified oceanic, Mediterranean, and southwest basin, based on the typology proposed by Joly et al., 2010). The
validation set merges six municipalities in the southwest basin climate setting, from the same classification. We choose
municipalities corresponding to different urban contexts (dense urban to rural) to have a representative sample as Corti et al.
(2011) showed that urban centers are not susceptible to subsidence, while discontinuous urban areas are half as vulnerable as
rural areas.

The locations of each set, as well as the climate (Joly et al., 2010) and exposure to clay shrink-swell maps (BRGM, 2023), are
detailed in Fig. 2. Each calibration subset is referred to as the corresponding French department (administrative subdivision):
Bouches-du-Rhône (13), Haute-Garonne (31), Loiret (45), Moselle (57) and Puy-de-Dôme (63).

## 3 Results

### 3.1 Assessment of the drought index configurations

For the calibration set consisting of the five regional subsets, the 1200 drought index configurations are evaluated by measuring
the Kendall rank correlation with the normalized number of claims. Figure 3 shows the Kendall tau coefficients averaged over
the calibration subsets and by 5 percentile threshold groups for each drought index type, simulation configuration, and model
soil layer. Each group, of equal size, consists of 25 values (5 calibration subsets x 5 thresholds). The maximum p-value obtained
per group is plotted in the Supplementary Material section.

Figure 3 shows that the average Kendall tau calculated over the groups ranges between 0.13 and 0.42. Similarly, the maximum
p-value observed per group ranges between 0.0001 and 1. For both LAI_model and LAI_clim simulations, and especially for



the duration index, the results tend to get worse as the thresholds get higher. We also note that the performance of the severity index decreases significantly for the most superficial 0.2-0.4 m model soil layer. The most stable index is the magnitude index. For extreme thresholds (percentiles 1 to 5 %), there is little difference between the three indices, and the average Kendall tau increases with the depth of the soil layer. Finally, the highest correlation coefficients are obtained with the LAI_model simulation, extreme thresholds and layers below 0.6 m (model soil layers 7 and 8). We also observe with the LAI_model

configuration that the p-value of each group is always below 0.001 for model soil layer 8 (0.8-1.0 m), cf. Figure S1 in the Supplement.

## 3.2 Optimal drought index

We define an optimal drought index from the LAI_model simulation, the SWI of the deepest model soil layer (layer 8, from 0.8 to 1.0 m), and averaged magnitudes computed with thresholds corresponding to percentiles 1 to 5 %. This index is hereafter

referred to as the optimal drought magnitude.

Figure 4 plots the average optimal drought magnitude of the five calibration subsets against the corresponding average normalized number of claims over the entire study period. The error bars indicate the amplitude of the values (minimum and maximum of the variable in the subset). The same data is also presented as a scatterplot in the Supplementary Material section. The CatNat decrees issued over the period are also shown in Fig. 4, as gray bars whose shading changes with the number of

decrees per year and subset. Over the whole period, we count 20, 25, 6, 12, and 16 decrees promulgated for subsets 13, 31, 45, 57 and 63, respectively.

From Fig. 4, we can see that the optimal drought index identifies 2003 and 2018 as the most important drought years for almost all subsets. This pattern appears in the normalized number of claims with some gaps. Discrepancies are also identified for years with zero drought magnitude when claims were reported, with the example of 2005 for Department 13. Some local

drought events are detected by both magnitude and insurance data, such as 2011 for Department 31. The amplitude of the magnitudes indicates that when a drought is detected, every municipality in the calibration subset is affected. This means that the index is spatially consistent. Overall, there is a good correspondence between zero magnitudes and the absence of damage. Furthermore, we find a very strong association between normalized damage counts and CatNat decrees.

The predicted counts and scores calculated for all subsets are shown in Table 2. For all calibration subsets, the PC and POD

are above 0.5, and the FAR and POFD are below 0.5 (except for the high FAR = 0.57 for zone 57). On the other hand, the obtained bias ranges between 0.64 and 1.75, with the subset having the smaller bias (closest to 1) being subset 45 (B = 1.18). The scores are averaged when the calibration subsets are merged, compensating for differences.

## 3.3 Validation

For the six municipalities of the validation set, we compute optimal drought magnitudes (LAI_model, layer 8, average of the

1st to 5th percentiles). The Kendall rank correlation between the optimal drought magnitude and the normalized number of claims yields a tau coefficient of 0.23 and a p-value less than 0.001, indicating a statistically significant correlation. Figure 5



plots the average optimal drought magnitude and the normalized number of claims for this set over the nineteen-year study period. CatNat decrees are shown as gray bars in the background of the second graph. These results are also presented in a scatterplot in the Supplementary Material section.

In Fig. 5, we see that the magnitude index is able to identify droughts in more than half of the municipalities for the years 2003, 2009, 2011, 2012, 2014, 2016 and 2018. However, not all of these years correspond to years with significant damage (2002, 2003, 2011, 2012, 2016, 2017), and some years even remain undetected. As for the calibration set, we observe a very strong correspondence between normalized damage numbers and CatNat decrees. The scores obtained for the validation set, displayed in Table 2, are quite similar to the ones obtained for the merged calibration subsets.

Then, the optimal magnitudes of calibration and validation sets are sorted into four classes. Statistics describing the normalized number of claims per magnitude class for both sets are available in Table 3. Figure 6 shows the distribution of the same variable per class, separating the two sets.

From Table 3 and Fig. 6, we can see that the entire population is evenly distributed among the classes for the two sets, except for classes 2 and 3 of the validation set, which concentrate more and less elements than class 1, respectively. For the calibration

set, the value of the damage quartiles increases with the magnitude class, indicating a positive association. The same is observed for the validation set for the lower and middle quartiles. We observe that the normalized number of claim distributions are similar for both sets in classes 0 and 2. Class 1 of the validation set is characterized by a high upper quartile and class 3 by a low one, in contrast to the wide range noted in class 3 for the calibration set.

## 4 Discussion

In the previous section, we presented an optimal drought index, that we calibrated over five regional subsets, and compared to independent data. The findings are discussed in the following section, along with perspectives.

### 4.1 How does the drought magnitude compare to existing drought indices?

This paper presents a new drought index, the yearly drought magnitude, which quantifies the yearly summed dry anomaly of an LSM-derived daily SWI. The reason for choosing a drought index based on LSM output is based on the nature of the

phenomenon under study. As mentioned above, clay shrinkage is a physical response of soils to drying. Therefore, the most relevant way to monitor it is to focus on soil moisture variations. Meteorological variables alone are not able to provide a sufficient description of the hydric state of soils due to the multiple interactions between atmosphere, soil and vegetation at the surface, hence the difference between SPI and SSWI noted by Soubeyroux et al. (2012) and Charpentier et al. (2022). As illustrated by the indices listed by WMO and GWP (2016), the only way to assess soil moisture evolution is to weight

contributions, from simple water balance models to more complex LSMs.

Although both are based on an ISBA-modeled SWI, our approach has a major difference from the one developed by Vidal et al. (2010). In the latter, the magnitude index is calculated from a SSWI consisting of a monthly SWI standardized at different




time scales by fitting to a given distribution over the period 1958-2008. This method was not applicable to our work because the insurance data leads us to focus on the years 2000-2018, which is too short for a proper distributional analysis of the

monthly SWI. However, the standardization step is still included in the index calculation since the thresholds are based on SWI percentiles.

## 4.2 Why is interactive LAI needed?

The evaluation of 1200 possible index configurations showed the best performance of the magnitudes computed from the LAI_model simulation, model soil layer 8 (0.8-1.0 m) and extreme drought thresholds (percentiles 1 to 5 %).

The only difference between LAI_model and LAI_clim simulations is that LAI varies over time in response to water and energy budgets for LAI_model, while the LAI cycle is the same each year, based on a climatology for LAI_clim. The interannual variability of LAI depends on meteorological conditions and the effect of soil water deficit on photosynthesis and plant growth. At the same time, the simulated soil moisture depends on the amount of water extracted from the soil by plant transpiration, and thus the temporal evolution of soil moisture is linked to the LAI variable. The interactive leaf area index

scheme of LAI_model is an improvement over the fixed pattern of LAI_clim because it includes vegetation feedback in response to soil drought. Drought conditions limit plant growth, LAI values, plant transpiration and root water extraction from the soil. Figure 7 shows time series of LAI, SWI5 and SWI8, for LAI_clim and LAI_model simulations at a single ISBA grid point located in the calibration subset corresponding to Department 31. It can be observed that larger LAI values in a simulation tend to trigger smaller SWI8 values at the end of the summer and during fall. For example, the lower soil moisture content of

LAI_clim in 2003 with respect to LAI_model corresponds to higher LAI values during the warm season. This effect is much more visible for SWI8 (0.8-1.0 m) than for SWI5 (0.2-0.4 m). We can explain this by the fact that deeper soil layers are more isolated from the surface than shallow ones, so changes in their moisture content are more dependent on root water extraction. Figure 3 shows that all annual drought indices correlate better with the number of insurance claims for deep soil layers. The best score values are obtained for SWI8. Figure 7 shows that the main difference between the two SWI variables is the presence

of a high frequency component in the SWI5 time series. The proximity of this most superficial layer to the surface, where meteorological exchanges take place, explains these short-term fluctuations, which consequently fade with depth. In other words, the SWI of a deep soil layer, which indirectly filters out high-frequency variations, reflects longer-term soil moisture trends than surface soil moisture. This explains the better correlation of the SWI8-based drought indices with damage data, considering that subsidence is a long-term, gradual phenomenon. For all simulations, layers and index types, better results are

observed for low thresholds. Consequently, a daily SWI below the 5th percentile of the empirical distribution is relevant for the definition of drought when monitoring subsidence.

## 4.3 Is the optimal drought magnitude a reliable proxy for subsidence damage?

Figure 3 shows that magnitude is a more robust drought index for representing subsidence damage than duration and severity. In particular, a deterioration in the performance of the duration and severity indices is observed for increasing percentile



drought thresholds and decreasing soil layer depths. The duration index loses much information for large percentile drought
thresholds, hence the lower correlation values. As with severity, this index is inherently sensitive to extreme values, which is
not well suited to the frequent soil moisture variations of near-surface layers that are not representative of long-term trends (as
seen in Fig. 7). Nevertheless, we note very similar Kendall tau values for the three indices at percentile drought thresholds less
than 15 %. Finally, magnitude is chosen as the best index because it combines both duration and severity, compensating for

their respective drawbacks. Also, as shown in Fig. 4, magnitudes are spatially consistent: similar drought trends are observed
for points within the same regional calibration subset. The average Kendall tau of 0.42 between magnitudes and the
standardized number of claims for the calibration set shows a strong positive association, with a high statistical significance
(p-value $\leqslant 0.001$).

The optimal magnitude index identifies significant drought events in 2003 and 2018 for more than half of the study areas (see

Fig. 4). These two years experienced notable summer drought events, which have been widely documented in the literature
(see for example Buras et al. (2020)). The index has the advantage of identifying not only the occurrence of such droughts, but
also their absence, and of being spatially consistent.

Nevertheless, we observe discrepancies: drought years with no damage recorded, and conversely, damage recorded in the
absence of drought. The functioning of the French CatNat system offers possible explanations for both situations. As explained

above, the coverage of subsidence damage by this system requires the publication of a decree for a given year and municipality,
the issuance of which is conditioned by geotechnical and meteorological criteria. The meteorological criterion is currently
calculated on a seasonal basis, and requires a frequency threshold to be exceeded. Losses are only documented in years with
decrees, hence their strong co-occurrence noted in Figs. 4 and 5. The absence of decrees provides an explanation for the
inconsistencies between positive magnitudes and zero damages. A drought event can occur without leading to a decree if the

seasonal criteria are not met. On the other hand, it is possible to have low drought magnitudes and a CatNat decree in the same
year. Low threshold magnitudes quantify the driest moments of the year (summer to autumn droughts), while a single season
only needs to be drier than usual to satisfy the CatNat meteorological criterion. For such years, the presence of claims can be
explained by delays in reporting, as it is difficult for both homeowners and insurers to accurately time the occurence of
subsidence damage. This would explain why, for certain years and municipalities, claims are reported without significant

magnitudes. The link to the CatNat regime is not the only factor explaining the inconsistencies between magnitudes and claims:
several sources of uncertainty have not been considered in this study. These are discussed in the next section.

The analysis of the scores provides more insight, subset-wise. The high PC ($> 0.5$) obtained for all zones shows that the index
makes a majority of correct predictions, despite the limitations mentioned above. The same conclusions can be drawn from
the POD ($> 0.5$): more than half of the loss events were detected, for each subset. On the other hand, the important range of

bias variation was unexpected: using thresholds defined from a given SWI frequency, the occurrence of subsidence damage is
either under-predicted (subset 31) or over-predicted (subset 57). We can link this finding to the record of CatNat decrees: for
the same number of years and cities, 25 decrees were issued for subset 31, and 12 for subset 57. Overprediction is more likely
for a small number of observations, and vice versa. This is not the only explanation, as the less biased subset (45, B = 1.18) is



associated with the smallest number of decrees (6). Finally, FAR and POFD characterize the tendency of the index to make
false predictions. For all subsets, the POFD is below 0.4: fewer than 4 out of 10 predictions turn out to be wrong in the absence
of subsidence claims. The FAR is also below 0.4, except for subset 57: fewer than 4 positive claims predictions out of 10 turn
out to be wrong. In both cases, a minority of predictions are wrong. The high FAR obtained for subset 57 (0.57) is related to
the high bias (1.75) noted above. The scores computed over the entire calibration data set compensate for the differences
between the subsets: these results, which are more robust, were expected, given the largest number of conditions.

The validation step also provides useful information to analyze how the optimal magnitude index fits to the subsidence hazard.
The lower Kendall tau of 0.23 indicates a weaker association of the index with subsidence damage than for the calibration set,
while maintaining high statistical significance (p-value less than 0.001). Once again, we can explain the disparities between
drought magnitude and damage by the absence of CatNat decrees and the delays in reporting. Nevertheless, the magnitude
index manages to identify most of the years with significant damage. The results obtained for the validation set from the
contingency table are quite similar, if not slightly less effective, than those obtained for the entire calibration set. The
satisfactory performance of the index computed from unseen data proves that we successfully avoid overfitting, i.e. the over-
adaptation of the model to specific patterns of a given data set, in our case the calibration set. In addition, the magnitude
classification reveals underlying trends in both the calibration and validation sets (Fig. 6). The similar distributions of both
sets across classes indicate that the validation set, although smaller, is representative. However, these distributions are uneven,
with the first class, corresponding to no damage, concentrating more than half of the magnitudes (Table 3). This is not
surprising since we are focusing on extreme and thus rare events. This is problematic for the validation set, as there is only a
limited amount of data left to divide into classes. For example, the last class in the validation set contains 9 % of the magnitudes,
or 10 values, which is too small for a proper analysis. While the data in the validation set are representative of the proportion
of null events, they are insufficient to provide reliable damage distributions associated with positive magnitudes. This provides
an explanation for the incoherent decreasing upper quartile observation from magnitude class 1 to 3 in Table 3. The zoning
between classes can be refined by using a larger data set. While the magnitude drought index is able to identify the occurrence
of drought events relevant for subsidence damage monitoring, the index alone is not able to provide accurate quantitative
damage estimates at this stage. The validation step could be improved by using a larger data set.

## 4.4 What are the sources of uncertainty?

The non-linearity of the relationship between optimal drought magnitudes and losses can be explained by several sources of
uncertainty. Subsidence losses are the result of a combination of factors, not all of which have been considered in the same
level of detail in this paper.

### 4.4.1 Representativeness of the insurance database

The number of subsidence claims used in this paper is a sample from a database compiled by CCR from insurance company
submissions. It is important to keep in mind that the representativeness of this data varies across years and cities, depending




on what was submitted. This is truer the smaller the city (see the number of single-family houses with moderate to high exposure to clay shrinkage per city in the Supplementary Material).

### 4.4.2 Circularity

As already explained, the knowledge of the history of the CatNat decrees issued is crucial for the interpretation of a subsidence
damage record. However, the meteorological criterion that conditions the issuance of these decrees is based on the ISBA model, just like the magnitude index we developed, although in a different configuration. Therefore, there is an indirect link between the two variables we are correlating, which induces a certain circularity in the approach. This weakness is recognized, but no valuable alternative can be considered: insurance claims are the only available evidence of subsidence, and LSMs are the tools that currently provide the most consistent estimates of soil moisture at the national scale. Nevertheless, we can
consider the dependency between magnitudes and claims as low, since the meteorological criterion of the CatNat decree is based on a version of the ISBA model (SWI uniforme) that is very different from the one used in this study and that has been modified several times.

### 4.4.3 Differences of temporal and spatial resolution

Uncertainty is also introduced by the different temporal and spatial resolution of the two main datasets compared in this study,
i.e. soil moisture and number of claims.

The annual temporal resolution of this study is forced by the insurance claims dataset, while the soil moisture simulations are more precise (daily). The coarse resolution of the first dataset is explained by the difficulty of insurers to precisely date the occurence of subsidence damage due to the slow nature of the phenomenon. For this reason, this work could not be carried out at a finer temporal scale. Nevertheless, there is an advantage to using daily soil moisture data: the SWI distribution is robust
over a 19-year period. This is especially important in the context of a changing climate.

The two datasets also present a difference in spatial resolution: the number of claims is available at the municipal scale, while soil moisture is modeled at an 8 km x 8 km scale. These two grids are of the same order of magnitude. The overlap allows them to be paired using a criterion other than distance alone.

### 4.4.4 Cumulative effect of subsidence

After a particularly hot summer, the soil under perennial plants is likely to be so dry that a single wet season only may not be able to compensate for the water deficit. This leads to a cumulative effect of droughts and thus soil movement (Page, 1998). For simplicity, the drought index described in this study is calculated from the SWI of a single year only. The cumulative effect is therefore neglected and is a source of uncertainty.





### 4.4.5 Zoning of clay shrink-swell exposure

Clay shrinkage depends on triggering factors, such as drought, and predisposing factors, the most important of which are the presence of clay, its nature, and its ability to shrink and swell. In order to minimize the influence of these factors, we focused our analysis on the housing stock located in moderate to high hazard zones of municipalities with a significant history of subsidence damage. The blending of moderate and high exposure zones in the housing dataset is a first source of uncertainty, as the same drought is expected to trigger different responses in soils of different susceptibility. Despite the selection of the

housing dataset, the uncertainty arising from the exposure to clay shrink-swell exposure map itself must be considered. The exposure map, made at a regional scale of 1:50000, does not integrate very local clayey soil occurrences. The latter are frequent, in relation to clay formation and depositional geological processes (BRGM, 2023). Moreover, this study does not take into account the intensity of the deformation induced by the shrink-swell mechanism, which is a significant an important component explaining the deformations of the structures induced by variations in water content.

### 455 4.4.6 Local factors

Local factors other than the sporadic presence of swelling clay play a significant role in the subsidence phenomenon:

- The presence of vegetation (especially deciduous trees) around the building is critical, as roots are known to locally intensify soil drying during droughts (Freeman et al., 1992; Hawkins, 2013; Page, 1998).
- The characteristics of the building itself (e.g. building type, number of floors, presence of a basement) determine its

vulnerability to subsidence damage. This is important in understanding the causes of a subsidence claim (Page, 1998).
- Various elements of the building environment, such as its orientation, the slope of the land, or the presence of pavement, can limit or enhance soil moisture variations and thus seasonal ground movements (Cooling and Ward, 1948; Page, 1998).

None of these factors were considered in this work because this information was not available for integrated municipal-scale

data. The specific effect of trees on soil moisture fluctuations could be investigated by modifying the ISBA model configuration. For all the other local factors, evaluation of their influence on subsidence is only possible through local analysis using instrumented sites.

### 4.5 What are the possible applications for this work, in and outside of France?

The main result of this research is a new index, the yearly drought magnitude, specifically adapted to the problem of clay

shrinkage-induced subsidence. It is computed from the SWI output of the ISBA LSM, and calibrated and validated with insurance data. As shown above, this index is relevant for the identification of subsidence-induced drought events. It faces several limitations due to the various sources of uncertainty.

The research was carried out on a sample of 20 cities, but the index can be calculated for the whole country. A first possible application is the use of the magnitude of drought itself to monitor conditions likely to cause subsidence over time, as is



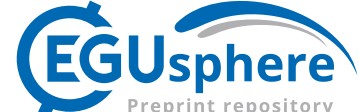

currently done by French institutions. Another possible application, this time for the insurance industry, would be to use the

index as an improved predictor in subsidence loss models, such as the one developed by Charpentier et al. (2022).

As several regions worldwide face similar clay shrinkage induced subsidence problems (see for example MacQueen et al. (2023) and Mostafiz et al. (2021) for the UK and the USA respectively), the methodology can be adapted to other countries. The SURFEX modeling platform, in which ISBA is implemented, can be applied anywhere, provided that high-resolution

atmospheric forcing is available.

## 5 Conclusion

In this paper, we propose a new annual drought index, the optimal drought magnitude, tailored to the subsidence hazard. Summary:

- The proposed optimal drought magnitude is the mean of five annual integrals of daily Soil Wetness Index (SWI)
values under several thresholds (percentiles 1 to 5 % of the empirical SWI distribution). It is based on the $8^{th}$ model layer (depth of 0.8 to 1.0 m) of the ISBA land surface model in a configuration that allows interactive LAI simulation. In this configuration, the average Kendall tau between the drought index and the normalized number of claims is equal to 0.42 with a high statistical significance (p-value $\leqslant$ 0.001). The validation step indicates that we avoid overfitting.

- The optimal drought magnitude identifies events that are likely to generate subsidence claims. It is spatially consistent. Differences with subsidence damage claims can be explained by the way claims are collected and by the lack of information on local conditions.

- The optimal drought magnitude benefits from recent advances in land surface modeling (multi-layer, interactive LAI). It could be used as a predictor in subsidence loss models to provide more accurate cost estimates. As this index is
based on ISBA simulations, future subsidence damage risks could be assessed by forcing ISBA through downscaled climate model simulations.

Future developments will focus on reducing uncertainties by working at finer spatial scales and by investigating the specific effects of different vegetation types.

## Code availability

The analysis was carried out with Python codes that can be made available upon request.

## Data availability

The data presented in the Figures are available online at: https://doi.org/10.6084/m9.figshare.23559507



**Supplement**

The supplement related to this article is available online at:

**Author contributions**

SBa and JCC designed the experiments. SBa performed the investigation, did the formal analysis and wrote the paper. BB processed the lai_model dataset. All co-authors participated in the interpretation of the results and the revision of the paper.

**Competing interests**

The contact author has declared that none of the authors has any competing interests.

**Acknowledgments**

The PhD thesis work of Sophie Barthélémy was co-funded by BRGM (Bureau de Recherches Géologiques et Minières), CCR (Caisse Centrale de Réassurance) and Météo-France. The authors would like to thank the operational services of Météo-France (DCSC), especially Jean-Marie Willemet, for providing the lai_clim soil moisture dataset.
The authors declare no conflict of interest.




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




**Table 1: Scores for assessing the performance of the optimal drought magnitude, based on the fractions of True Positives, True Negatives, False Positives, and False Negatives (TP, TN, FP, FN, respectively).**

| Name | Equation | Optimal value | Definition |
|---|---|---|---|
| Proportion Correct | PC = (TP+TN)/(TP+TN+FP+FN) | 1 | Ability to predict correctly (accuracy) |
| Bias | B = (TP+FP)/(TP+FN) | 1 (unbiased) | Tendency to see more (>1) or less (<1) predictions than observations |
| False Alarm Ratio | FAR = FP/(TP+FP) | 0 | Fraction of predictions that turn out to be wrong |
| Probability Of Detection | POD = TP/(TP+FN) | 1 | Fraction of occurrences of the event for which there was a prediction |
| Probability Of False Detection | POFD = FP/(FP+TN) | 0 | Fraction of wrong predictions when the event did not occur |

**Table 2: Table of contingencies (TP, TN, FP, FN) and scores (PC, B, FAR, POD, POFD) for calibration and validation subsets. Largest and lowest values are in bold and in italics, respectively.**

| Subset | | Number | TP (%) | TN (%) | FP (%) | FN (%) | PC | B | FAR | POD | POFD |
|---|---|---|---|---|---|---|---|---|---|---|---|
| Calibration | 13 | 76 | 34 | 33 | *8* | 25 | 0.67 | 0.71 | *0.19* | 0.58 | *0.19* |
| | 31 | 76 | **40** | *14* | 9 | **37** | *0.54* | *0.64* | *0.19* | *0.52* | 0.39 |
| | 45 | 76 | 21 | **58** | 13 | 8 | **0.79** | 1.18 | 0.38 | 0.73 | *0.19* |
| | 57 | 76 | *20* | 47 | **26** | *7* | 0.67 | **1.75** | **0.57** | **0.75** | **0.36** |
| | 63 | 76 | 27 | 38 | 13 | 22 | 0.64 | 0.81 | 0.33 | 0.54 | 0.26 |
| | All | 380 | 28 | 38 | 14 | 20 | 0.66 | 0.88 | 0.33 | 0.59 | 0.27 |
| Validation | All | 114 | 29 | 28 | 16 | 27 | 0.57 | 0.80 | 0.35 | *0.52* | 0.36 |



**Table 3: Statistical distribution of normalized number of claims (in %) by magnitude class of calibration and validation sets. Class limits, number of observations and quartiles (Q) are indicated.**

| Magnitude class | Range | Calibration | | | | Validation | | | |
|---|---|---|---|---|---|---|---|---|---|
| | | **Number** | **Q25%** | **Q50%** | **Q75%** | **Number** | **Q25%** | **Q50%** | **Q75%** |
| 0 | 0 | 220 (58%) | 0.0 | 0.0 | 0.0 | 63 (55%) | 0.0 | 0.0 | 0.1 |
| 1 | (0, 0.03] | 54 (14%) | 0.0 | 0.0 | 0.1 | 18 (16%) | 0.0 | 0.1 | 1.0 |
| 2 | (0.03, 0.26] | 53 (14%) | 0.0 | 0.1 | 0.7 | 23 (20%) | 0.0 | 0.2 | 0.9 |
| 3 | (0.26, 2.81] | 53 (14%) | 0.1 | 1.0 | 3.8 | 10 (9%) | 0.0 | 0.5 | 0.6 |





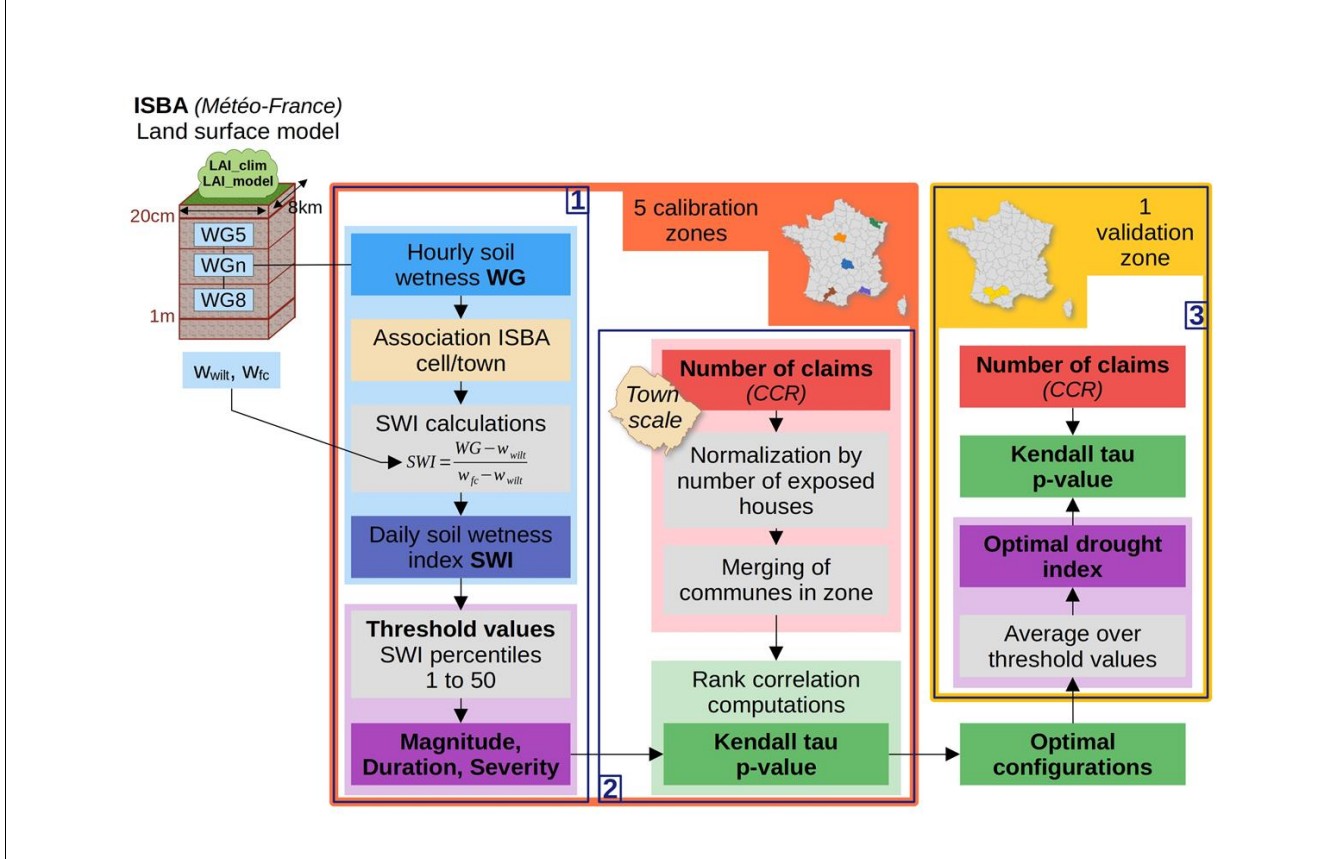

**Figure 1: Flowchart of the methodology deployed in this study with focus on the three main steps**

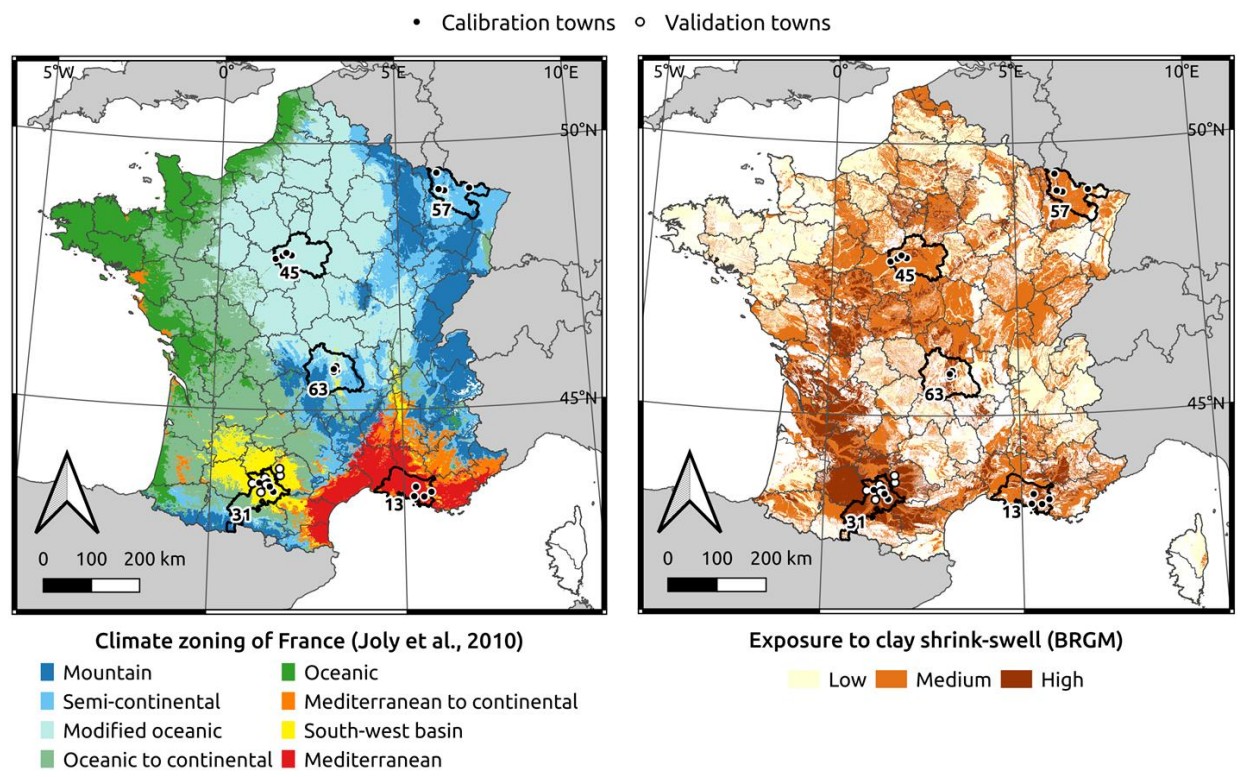

**Figure 2: Location of the 26 study municipalities, along with a climate (Joly et al., 2010) and clay shrink-swell exposure map (BRGM, 2019). The black markers depict the municipalities used for calibrating the indices, and the whites the ones used for validation. Bold lines indicate the five French departments (13, 31, 45, 57, 63) in which are located the calibration subsets (Bouches du Rhône, Haute-Garonne, Loiret, Moselle, Puy-de-Dôme, respectively).**



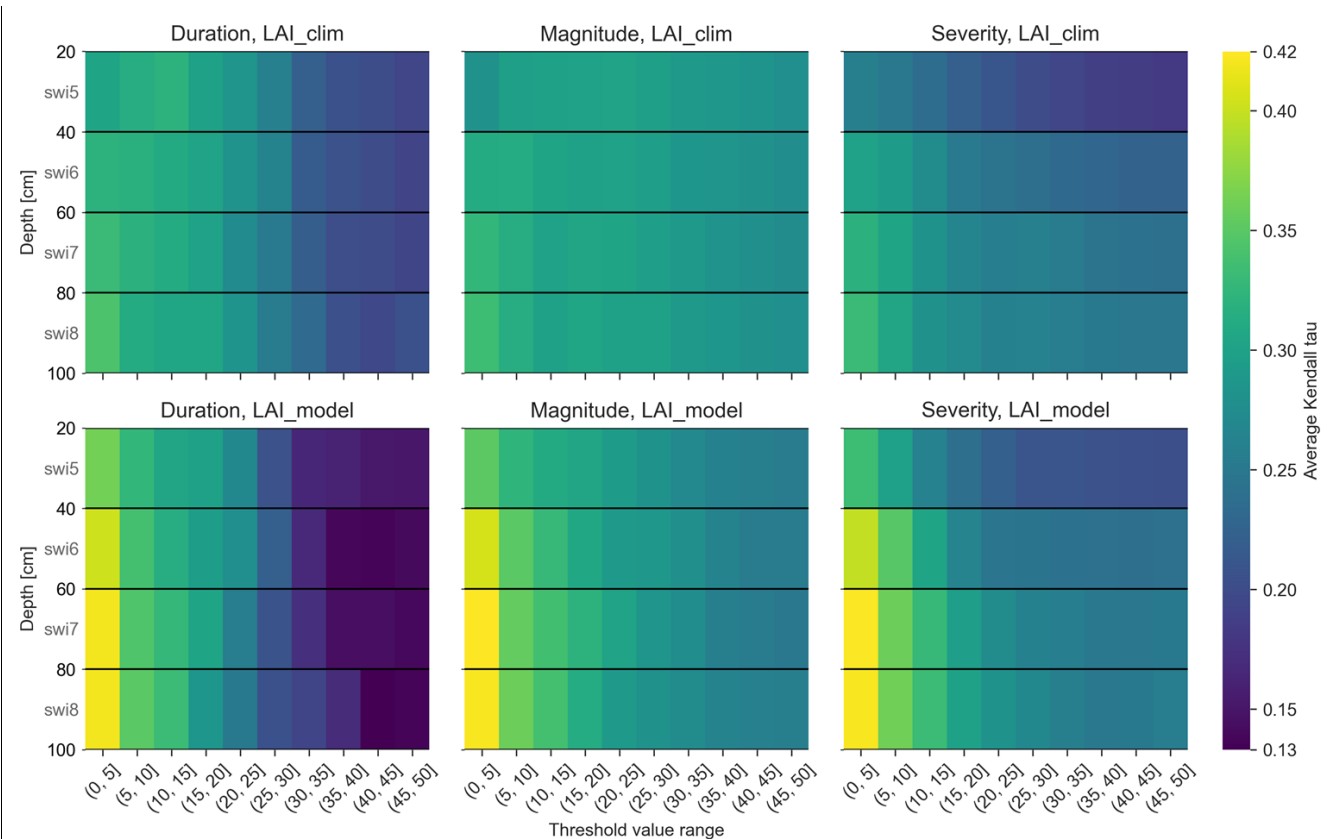

**Figure 3: Average Kendall tau per group, computing the rank correlation between drought index and normalized number of claims, for all calibration subsets, separating index type (duration, magnitude, severity), model simulation (LAI_clim, LAI_model), model layer (SWI5 to SWI8, i.e. 0.2-0.4 m to 0.8-1.0 m) and threshold value range (from 1 to 50 % percentiles in groups of five).**



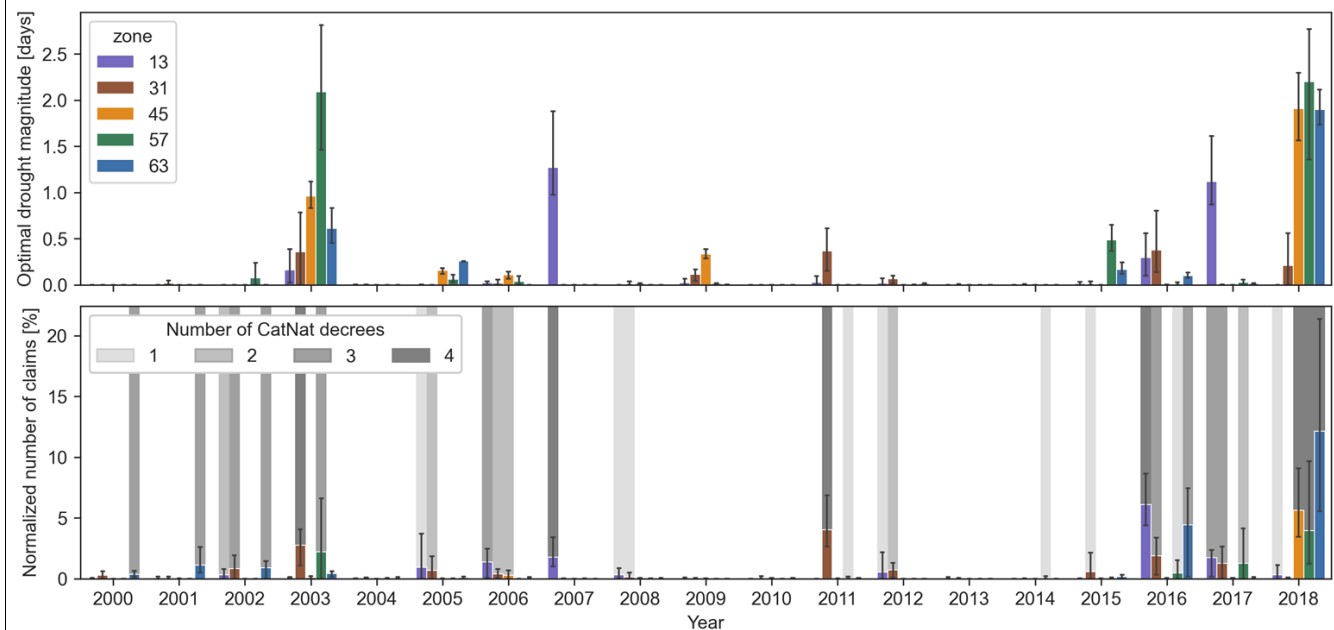

**Figure 4: Average optimal drought magnitude (top), and normalized number of claims and frequency of CatNat decree averaged by calibration subset (bottom). The error bars indicate the amplitude of values.**



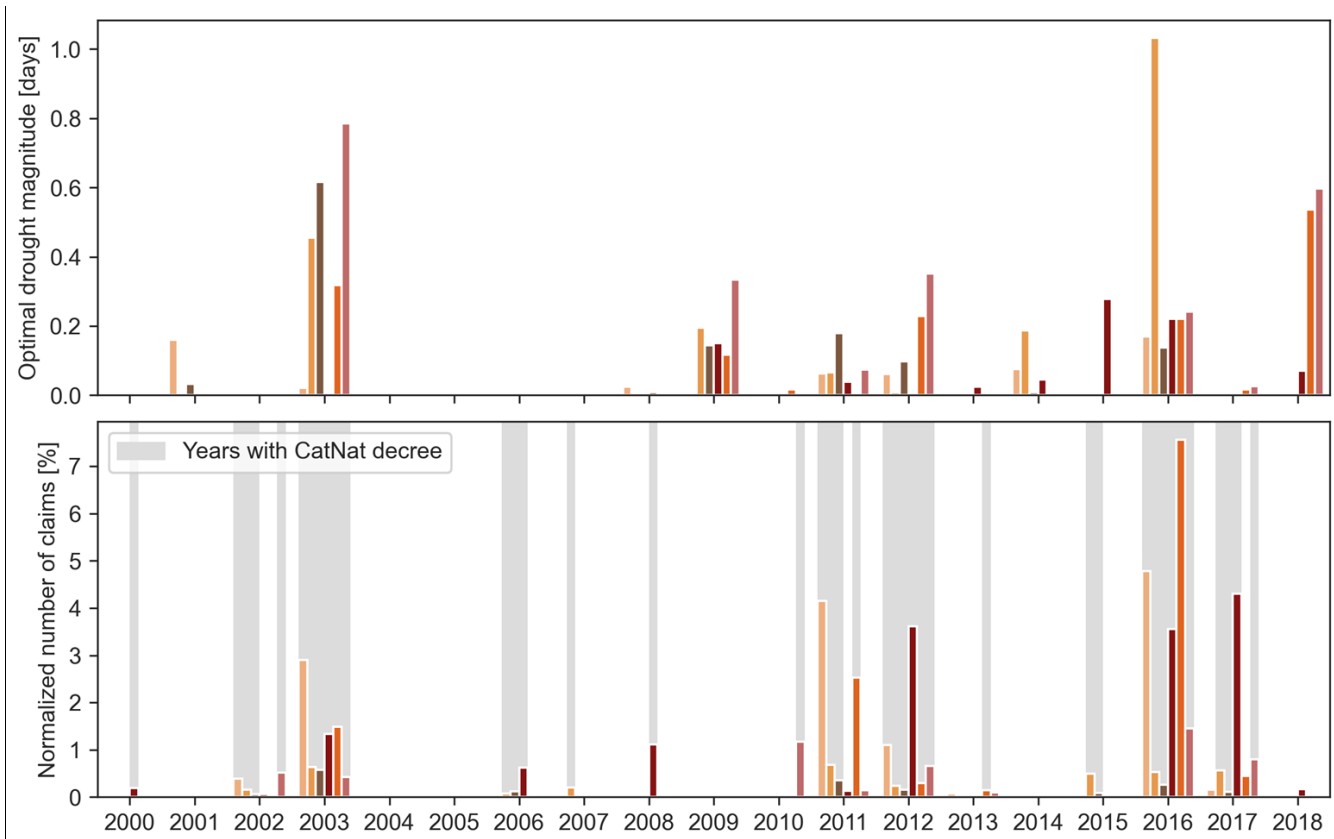

**Figure 5: Optimal drought magnitude (top), and normalized number of claims and CatNat decrees (bottom), for the six municipalities of the validation set. The colors correspond to the six different municipalities.**



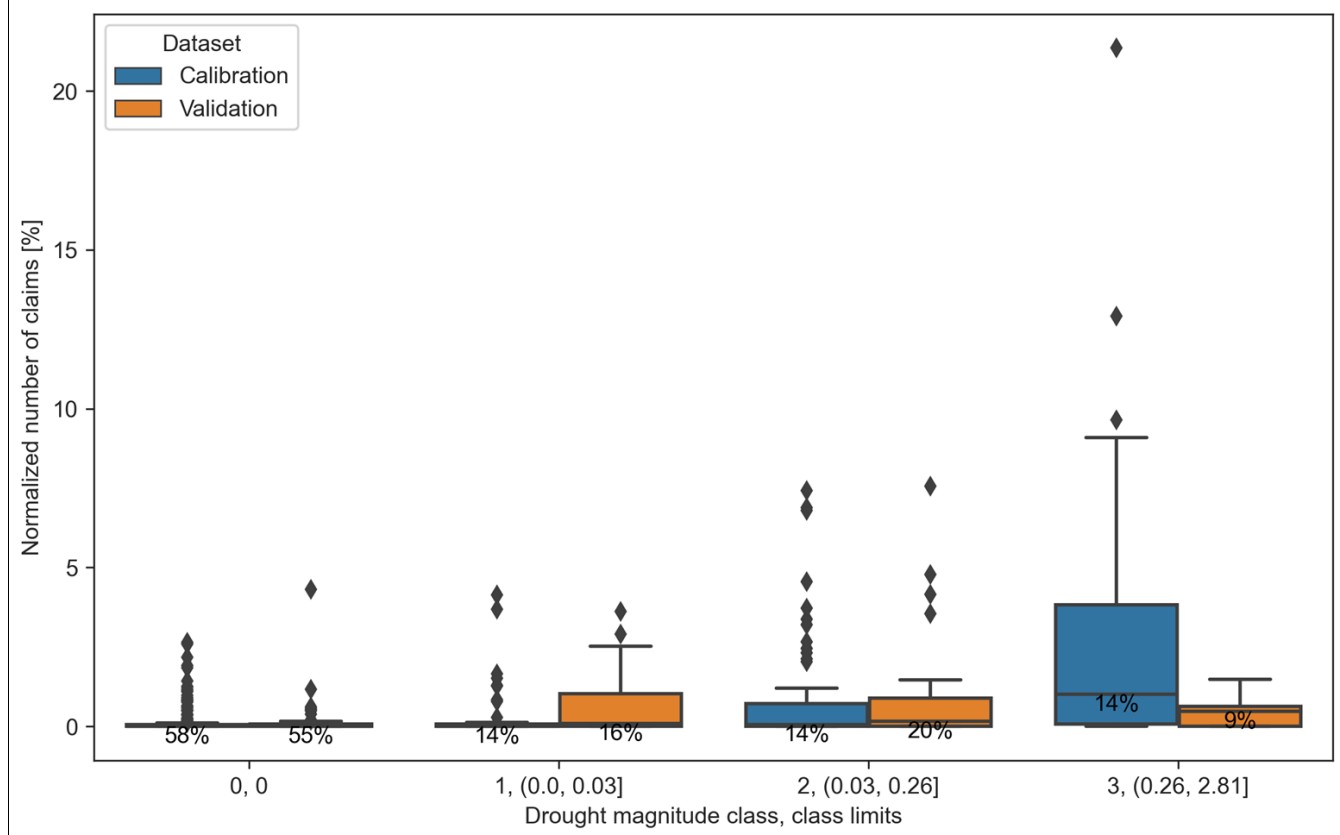

**Figure 6: Distributions of the normalized number of claims per magnitude class, separating calibration and validation set.**





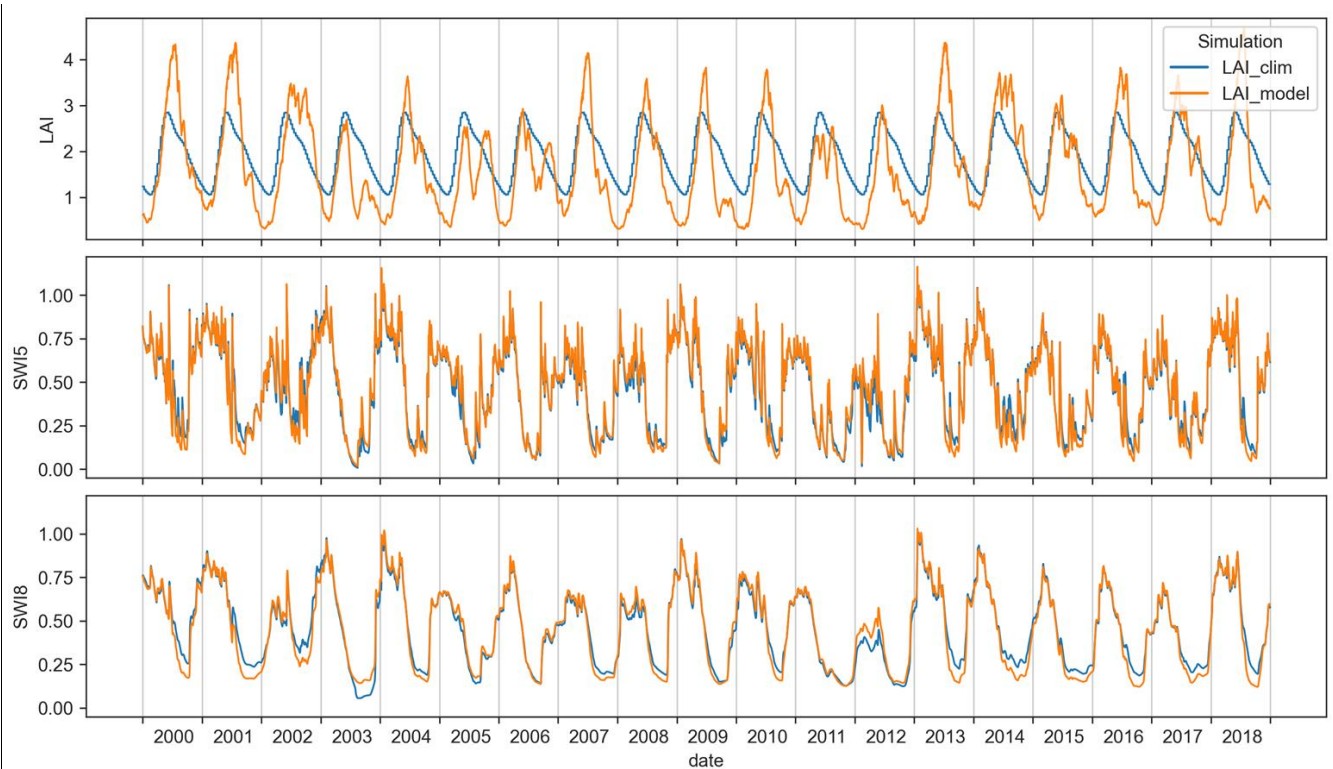

**Figure 7: Timeseries of LAI, SWI5 and SWI8 (corresponding to soil layers 0.2-0.4 m and 0.8-1.0 m, respectively), differentiating simulations LAI_clim and LAI_model, for a single ISBA grid point located in the calibration subset attached to department 31.**