# Peer review of "A new drought index fitted to clay shrinkage induced subsidence over France: benefits of interactive leaf area index"

_EGUsphere, 2023_

## Author Comment (AC3)

**Reviewer 3**

*The paper deals with an interesting numerical approach to calculate a drought index fitted to clay shrinkage induced subsidence over France. The reviewer asks the authors to give first more attention to the following general remarks in order to correct them.*

The authors thank the Anonymous Referee #3 for their constructive feedback on the work.

**COMMENT 3.1**: *I'm not really convicted that we can talk about a "new drought index" in this paper and choose it as a title! This can be misleading. The reuse of existing parameter SWI derived from ISBA simulation method, with an extended vegetation representation, is in my opinion not enough to name this parameter new drought index. I suggest to the authors to modify the title according to a "new approach" or an "adaptation" of an existing parameter.*

**RESPONSE 3.1**: We suggest modifying the title as follows:

> 'A new **approach for a** drought index fitted to clay shrinkage induced subsidence over France: benefits of interactive leaf area index'

**COMMENT 3.2**: *Add a legend for Figure 5 and specify the correspondence of each color bar.*

**RESPONSE 3.2**: Figure 5 was corrected as requested:

[Figure]

*Figure 5: Optimal drought magnitude (top), and normalized number of claims superposed to accepted and denied CatNat requests (bottom), for the six towns of the validation set.*

**COMMENT 3.3**: *page 4 and lines 100-101, moisture variations depend also on the mineralogy of clays and their saturated and unsaturated hydraulic conductivity, the initial soil suction and how water flows depending on its hydromechanical properties. Can the authors give more details on the choice of not taking soil parameters and behaviour into account in this study?*

**RESPONSE 3.3**: Soil parameters and behavior are accounted for indirectly in this study by the way the ISBA model works. As explained in Decharme et al. (2011), soil hydraulic parameters such as porosity, matric potential at saturation and saturated hydraulic conductivity are related in the model to soil texture through empirical pedotransfer functions (more specifically, Noilhan and Lacarrère (1995) relationships derived from Clapp and Hornberger (1978)).

We propose adding these elements to section 2.1.

**COMMENT 3.4**: *page 4 and lines 102-103, in the ISBA model, it is considered that texture is homogeneous and represented by some clay, sand and silt contents. This cannot reflect the reality when we know the heterogeneity of clayey soils in France, at the kilometer resolution and including at the same plot of the house. Thus, calculations made and improved based on the ISBA model and derived versions is a tool to have an idea to estimate the top surface soil moisture but it is still complex to deduce any real state of hydromechanical behaviour of clayey soils without considering their mineralogy, heterogeneity and hydromechanical properties such as soil water characteristic curve (SWCC).*

**RESPONSE 3.4**: As explained in response to Comment 3.3, soil hydraulic parameters are derived from texture in ISBA using empirical pedotransfer functions. As the reviewer correctly points out, it is true that texture averaged at the grid scale is not representative of what can be found on a house plot. Inferring hydromechanical behavior from this information alone would indeed be problematic. It is not the purpose of the drought index we are developing here. We suggest adding this point to the text of the revised manuscript.

**COMMENT 3.5**: *page 4 and lines 111-113, analysis of this study were based on four model layers until 1.0 m depth. One of the direct consequences of climate change is the propagation of soil desiccation in depth under severe and recurrent drought. This can reach 3.0 m depth and more depending on the close environment configuration. It would be interesting if the authors try to take into account this climate change effect through new calculations.*

**RESPONSE 3.5**: The reviewer has a good point. In this study we are limited to a maximum depth of 1m, due to the patch continuity requirement. Knowing that the amplitude of soil moisture variations decreases with depth (Ravina, 1983), a drought that reaches deep soil layers will intensively dry out shallow layers. Theoretically, such an event can be detected with surface layer information alone. It should be noted, however, that drying out occurs with a time lag increasing with the depth of the layer. We have verified in this analysis that even for the deepest model layers, the drought observed in a given year never overlaps with the following year, which would distort the index calculations. This assumption may be questioned under future climatic conditions, considering that the frequency and intensity of droughts in France will increase.

We suggest adding this comment to section 2.1.

**COMMENT 3.6**: *page 5 and line 127, what do the authors mean by "volumetric soil moisture"? Is it possible to explain how simulation can provide this physical property of the soil?*

**RESPONSE 3.6**: Volumetric soil moisture is the water content of the soil, expressed in units of volume of water per unit of volume of soil (m³/m³). As explained in the text, ISBA simulates this variable by performing a water balance between precipitation, drainage, evapotranspiration, runoff, and storage. We will add this clarification to section 2.1.

**COMMENT 3.7**: *page 5 and lines 129-131, can the authors clarify better the "conversion" of the volumetric soil moisture to soil wetness indices (SWI) to justify considering a single definition of drought?*

**RESPONSE 3.7**: The Soil Wetness Index SWI consists of the soil moisture normalized between the field capacity $W_{fc}$ and the wilting point $W_{wilt}$, as expressed in the equation below. The latter two hydraulic parameters are derived from the texture using pedotransfer equations.

$$SWI = \frac{WG - W_{wilt}}{W_{fc} - W_{wilt}}$$

To justify our approach, we propose to include the following figure in the supplementary material of the paper. The maps show (1) the soil texture defined in the model as (a) clay and (b) sand fractions, and (2) the volumetric soil moisture WG and (3) the soil wetness index SWI of layer 8 of patch 4, at the same time (16/08/2022, 10h). Volumetric soil moisture is highly dependent on texture (lower moisture content in sand than clay), making it impossible to use a single definition of drought for the whole country. Switching to SWI allows us to remove the influence of texture and characterize only temporal variations.

[Figure]

**COMMENT 3.8**: *page 6 and lines 179-181, it appears that this study is mainly based on SWI outputs of the ISBA model. I'm not convicted that these calculations are the most reliable tools for studying soil moisture variations as mentioned.*

**RESPONSE 3.8**: Land Surface Models that simulate soil moisture variables are reliable tools for evaluating changes on a large scale. As explained in response to Comment 3.7, the conversion to SWI enables to solely focus on variations by removing the dependence on soil texture.

Although the conversion is useful for unifying the information on a national scale, it does not affect the magnitude calculations. In fact, the calculations are based on thresholds defined by percentiles of the daily soil moisture, the distribution of which is unchanged by the linear transformation that is the conversion to SWI.

**COMMENT 3.9**: *page 6 and lines 189-190, I'm not sure that it is possible to assume that results based on in situ observations in the USA and Canada can be applicable to France especially under climate change context. Many assumptions are considered in this study, which show the complexity to approach the soil water content and its variations without taking into account its hydromechanical properties at a given initial state.*

**RESPONSE 3.9**: We agree with the reviewer's comment. We are conscious that the hypothesis of a stable daily soil moisture distribution is questionable under a climate change context. We propose stating this in the discussion part dealing with uncertainties, section 4.4.3:

> [L434] 'Nevertheless, there is an advantage to using daily **instead of annual** soil moisture data: the SWI distribution is **expected to be** more robust over a 19-year period**, due to the high number of observations.** This is especially important in the context of a changing climate. **The hypothesis of a stable daily soil moisture distribution remains a source of uncertainty inherent to this work.**'

**Additional references**

Burnol, A., Foumelis, M., Gourdier, S., Deparis, J., and Raucoules, D.: Monitoring of expansive clays over drought-rewetting cycles using satellite remote sensing, Atmosphere, 12, https://doi.org/10.3390/atmos12101262, 2021.

Clapp, R. B. and Hornberger, G. M.: Empirical equations for some soil hydraulic properties, Water Resour. Res., 14, 601–604, https://doi.org/10.1029/WR014i004p00601, 1978.

Crosetto, M., Solari, L., Balasis-Levinsen, J., Bateson, L., Casagli, N., Frei, M., Oyen, A., Moldestad, D. A., and Mróz, M.: Deformation monitoring at european scale: The copernicus ground motion service, in: International Archives of the Photogrammetry, Remote Sensing and Spatial Information Sciences - ISPRS Archives, 141–146, https://doi.org/10.5194/isprs-archives-XLIII-B3-2021-141-2021, 2021.

Decharme, B., Boone, A., Delire, C., and Noilhan, J.: Local evaluation of the Interaction between Soil Biosphere Atmosphere soil multilayer diffusion scheme using four pedotransfer functions, J. Geophys. Res. Atmospheres, 116, 1–29, https://doi.org/10.1029/2011JD016002, 2011.

Meisina, C., Zucca, F., Fossati, D., Ceriani, M., and Allievi, J.: Ground deformation monitoring by using the Permanent Scatterers Technique: The example of the Oltrepo Pavese (Lombardia, Italy), Eng. Geol., 88, 240–259, https://doi.org/10.1016/j.enggeo.2006.09.010, 2006.

Noilhan, J. and Lacarrère, P.: GCM Grid-Scale Evaporation from Mesoscale Modeling, J. Clim., 8, 206–223, https://doi.org/10.1175/1520-0442(1995)008<0206:GGSEFM>2.0.CO;2, 1995.

Ravina, I.: The influence of vegetation on moisture and volume changes, Géotechnique, 33, 151–157, https://doi.org/10.1680/geot.1983.33.2.151, 1983.

Tzampoglou, P., Loukidis, D., and Koulermou, N.: Seasonal Ground Movement Due to Swelling/Shrinkage of Nicosia Marl, Remote Sens., 14, 1440, https://doi.org/10.3390/rs14061440, 2022.

---

## Author Response (AR1)

**Reviewer 1**

The authors would like to thank the Anonymous Referee #1 for their valuable feedback on the submitted manuscript.

*COMMENT 1.1: The paper is very interesting, the topic is important, and the methodology considered is appropriate. My main concern is about the data used here, partially described in section 2.2. As explained in https://doi.org/10.5194/nhess-22-2401-2022 (Charpentier, James and Ali 2022) on a similar topic, the French system has a very specific design, where claims within a "town" (or "commune" or "municipality") need first a national recognition before beeing accepted as "legitimate claims" (and then paid by the insurance company). In Charpentier, James and Ali (2022), it is observed that models are good to predict town that will claim losses, but the national recognition stage is much harder. Which data are used in this study? Those obtained initially, from towns claiming losses, or those obtained after censoring, by national recognition? In the first case, the paper is ok, and could be published. Otherwise, there is a major selection bias in the study that should, somehow, be considered.*

RESPONSE 1.1: Many thanks for noting this. The insurance dataset used in this study corresponds to the accepted "legitimate claims", after the national recognition step (accepted CatNat requests). To investigate the influence of this national recognition stage on the end-of-chain insurance data, we propose to add to the paper an interpretation of the history of accepted and refused national recognition requests for the towns forming our sample (this data was obtained by merging individual decree files downloaded from the CCR website: https://catastrophes-naturelles.ccr.fr/les-arretes). For each year and subset, we confront the number of accepted and denied requests to the drought index and the reported claims. These elements are added to Figure 4 (see the new Figure below). The number of refused decrees is significant and does have an influence on the insurance data. We could identify situations in calibration and validation set where this bias could be the source of inconsistencies between drought index and claims. In particular, we can explain all the inconsistencies noted between index and claims (positive index and no claims) in 2003 and 2018 by this factor (3 inconsistencies in 2003 and 1 in 2018, in the Figure below).

The following text was added:

- Section 2.2: "**Subsidence claims correspond to accepted "legitimate claims", after a national recognition step of CatNat requests**" ; "**To investigate the influence of this national recognition stage on the end-of-chain insurance data, we also used the history of accepted and refused national recognition requests for the towns forming our sample. This data was obtained by merging individual decree files downloaded from the CCR website (**https://catastrophes-naturelles.ccr.fr/les-arretes**).**"

- Section 3.2: "**The number of rejected claims is also shown in Fig. 4. Situations in the calibration and validation sets can be identified where rejected claims could be the source of inconsistencies between the drought index and claims. In particular, all the inconsistencies noted between index and claims (positive index and no claims) observed in 2003 and 2018 can be explained by this factor (3 inconsistencies in 2003 and 1 in 2018)**."

[Figure]

*Figure 4 : Optimal drought magnitude (a), and normalized number of claims (b) averaged by calibration subset, and numbers of towns per subset (c) with accepted (A) or denied (D) CatNat requests. The error bars indicate the amplitude of values.*

**Reviewer 2**

The authors thank the Anonymous Referee #2 for their constructive feedback on the work.

**COMMENT 2.1**: *Check spelling but in line 139*

**RESPONSE 2.1**: The spelling error was fixed: 'butt' was replaced by 'but'

**COMMENT 2.2**: *Section 4.2. lines 332 the naming conventions are very confusing - can't you just refer to depths? For example why do you refer to the surface to 20cm as LAI and deeper as SW?*

**RESPONSE 2.2**: The naming conventions were clarified. LAI refers to the Leaf Area Index variable, whereas SWn refer to the Soil Wetness of layer n (for instance, SWI5 for layer 5 between 20 and 40cm and SWI8 for layer 8 between 80 and 100cm).

- Section 4.2: "Figure 7 shows time series of LAI, SWI5 and SWI8, for LAI_clim and LAI_model simulations at a single ISBA grid point located in the calibration subset corresponding to Department 31." was replaced by "**Figure 7 shows time series of LAI, SWI5 (0.2-0.4m) and SWI8 (0.8-1.0m). Time series are shown for LAI_clim and LAI_model simulations at a single ISBA grid point located in the calibration subset corresponding to Department 31.**"

**COMMENT 2.3**: *Do you mean December? Year - line 333 Section 4.2.*

**RESPONSE 2.3**: We do not understand the reviewer's comment on this particular line.

**COMMENT 2.4**: *Section 4.2 lines 336 - 338 support with evidence from the literature.*

**RESPONSE 2.4**: To justify the greater influence of root water extraction on soil moisture in deeper layers than in surface layers, we did the following modification to the original text:

'This effect is much more visible for SWI8 (0.8-1.0 m) than for SWI5 (0.2-0.4 m).  **As explained by Ravina (1983), the hydraulic conductivity of the top soil layer decreases with drying to the point where moisture in the deeper layers can remain practically unchanged. Soil moisture variations in deep layers are therefore more dependent on water uptake by roots than on diffusion processes. This explains the large impact of vegetation transpiration and the stronger correlation with LAI.**'

**COMMENT 2.5**: *In Section 4.4.2 line 423 you mention that household claims are the only available evidence of subsidence. Might you consider other sources such as InSAR which should work at the scale of postcode...we use this to monitor, for example, subsidence from mining operations.*

**RESPONSE 2.5**: We agree. We added to the 4.4.2 discussion section a paragraph developing the possible contribution of remotely-sensed vertical displacements. These techniques are used to track displacements over large areas, and are applicable to clay shrink-swell monitoring. However, such data was not available at the time of the study:

> '**A possible alternative to insurance claims as a proxy for subsidence is the direct use of remotely sensed ground motion. In particular, satellite-borne interferometric synthetic aperture radar (InSAR) data can be used to infer vertical motion after appropriate processing, as done by Burnol et al. (2021). For example, the European Ground Motion Service (Crosetto et al., 2021) provides vertical displacements over Europe with high spatial and temporal resolution, based on the Copernicus Sentinel-1 satellites, since 2018. The main advantage of this technique is its large spatial coverage. However, the interpretation of such data is not trivial. In the case of clay shrink-swell, the vertical displacements are non-linear (seasonal periodicity), of small amplitude (a few to tens of mm), and spatially heterogeneous, both due to the natural irregularity of clay soils and to the contrasting responses of reflectors (less movement is expected for tall buildings on pile foundations, as explained by Tzampoglou et al. (2022)). It can therefore be challenging to separate a signature expansive soil signal from other phenomena such as subsidence induced by water pumping (Meisina et al., 2006). We recognize the potential of these data, but the EGMS dataset was not available at the time of the study. In addition, it begins in 2018, which barely overlaps with our study period, which spans from 2000 to 2018.**'

**COMMENT 2.6**: *Section 4.4.4. This is a valid and important observation i.e. the claim may be made years after the problem started to occur. To put it slightly differently, the damage may the result of a cumulation of years of movement (shrink swell) in the soil or it may be the result of a once off event. Perhaps you can support this discussion point with some further references supporting your choice of one year timescale OR giving us a better idea of what the uncertainty may look like.*

**RESPONSE 2.6**: The reviewer asks us to justify our decision to base the drought index on data from a single year, when subsidence is known to be a cumulative problem. To address this point, we added the following text to the concerned paragraph:

> 'The cumulative effect is therefore neglected and is a source of uncertainty. **The good agreement obtained here between drought magnitudes and normalized claims indicates that the conditions of a single year are a satisfactory enough predictor of subsidence occurrence. Taking into account the cumulative effect would improve the agreement with the numbers of claims. This step could be implemented in a damage model by, for example, weighting magnitudes by their history.**'

**COMMENT 2.7**: *Section 4.4.5. lines 449 - 454 - Can you suggest how this problem might be overcome?*

**RESPONSE 2.7**: In the article, we identify the resolution of the clay shrink-swell hazard zoning map as a source of uncertainty. The reviewer asks us to mention solutions to overcome this problem. We added the following text to the existing paragraph in Section 4.4.5:

*"The lack of precision of the clay maps here affects the number of houses in different hazard zones used in the normalization step. The associated uncertainty is transferred to the value of the normalized number of claims. At this stage, we are not trying to make precise damage predictions, only to identify drought years. Therefore, this source of uncertainty is not the most dominant."*

**COMMENT 2.8**: *Overall comment: This may be a slightly naive question, but would it be possible to validate the model by comparing to locations where you have subsidence data - or even cross reference with InSar data? You are basically using the claims data as a proxy for subsidence, as pointed out earlier, there may be other sources of data both point and remote sensing data that is publically available, that can be used.*

**RESPONSE 2.8**: Thank you for this comment. As detailed in response to comment 2.5, InSAR data have potential but were not available at the time of the study, and do not cover the whole period. As for the use of point-data, we do not have and are not aware of any openly-available insurance damage data at a finer spatial scale than the town.

**Reviewer 3**

*The paper deals with an interesting numerical approach to calculate a drought index fitted to clay shrinkage induced subsidence over France. The reviewer asks the authors to give first more attention to the following general remarks in order to correct them.*

The authors thank the Anonymous Referee #3 for their constructive feedback on the work.

**COMMENT 3.1**: *I'm not really convicted that we can talk about a "new drought index" in this paper and choose it as a title! This can be misleading. The reuse of existing parameter SWI derived from ISBA simulation method, with an extended vegetation representation, is in my opinion not enough to name this parameter new drought index. I suggest to the authors to modify the title according to a "new approach" or an "adaptation" of an existing parameter.*

**RESPONSE 3.1**: We modified the title as follows:

> '**A new approach for drought index adjustment to clay shrinkage-induced subsidence over France: advantages of the interactive leaf area index**'

**COMMENT 3.2**: *Add a legend for Figure 5 and specify the correspondence of each color bar.*

**RESPONSE 3.2**: Figure 5 was corrected as requested:

[Figure]

*Figure 5: Optimal drought magnitude (top), and normalized number of claims superposed to accepted and denied CatNat requests (bottom), for the six towns of the validation set.*

**COMMENT 3.3**: *page 4 and lines 100-101, moisture variations depend also on the mineralogy of clays and their saturated and unsaturated hydraulic conductivity, the initial soil suction and how water flows depending on its hydromechanical properties. Can the authors give more details on the choice of not taking soil parameters and behaviour into account in this study?*

**RESPONSE 3.3**: Soil parameters and behavior are accounted for indirectly in this study by the way the ISBA model works. As explained in Decharme et al. (2011), soil hydraulic parameters such as porosity, matric potential at saturation and saturated hydraulic conductivity are related in the model to soil texture through empirical pedotransfer functions (more specifically, Noilhan and Lacarrère (1995) relationships derived from Clapp and Hornberger (1978)).

We added these elements to section 2.1:

"**As explained in Decharme et al. (2011), soil water holding capacity and soil hydraulic parameters such as porosity, matric potential at saturation, and saturated hydraulic conductivity are related to soil texture in the model through empirical pedotransfer functions. The latter are described in Noilhan and Lacarrère (1995) and derived from Clapp and Hornberger (1978).**"

**COMMENT 3.4**: *page 4 and lines 102-103, in the ISBA model, it is considered that texture is homogeneous and represented by some clay, sand and silt contents. This cannot reflect the reality when we know the heterogeneity of clayey soils in France, at the kilometer resolution and including at the same plot of the house. Thus, calculations made and improved based on the ISBA model and derived versions is a tool to have an idea to estimate the top surface soil moisture but it is still complex to deduce any real state of hydromechanical behaviour of clayey soils without considering their mineralogy, heterogeneity and hydromechanical properties such as soil water characteristic curve (SWCC).*

**RESPONSE 3.4**: As explained in response to Comment 3.3, soil hydraulic parameters are derived from texture in ISBA using empirical pedotransfer functions. As the reviewer correctly points out, it is true that texture averaged at the grid scale is not representative of what can be found on a house plot. Inferring hydromechanical behavior from this information alone would indeed be problematic. It is not the purpose of the drought index we are developing here. We added this point to Section 2.1:

"**Because soil texture is averaged within a model grid cell, this approach provides a large scale representation, but is not representative of what may be found on a house lot. Inferring the hydromechanical behavior of clayey soils from this information alone would be problematic. This is not the purpose of the drought index developed in this study.**"

**COMMENT 3.5**: *page 4 and lines 111-113, analysis of this study were based on four model layers until 1.0 m depth. One of the direct consequences of climate change is the propagation of soil desiccation in depth under severe and recurrent drought. This can reach 3.0 m depth and more depending on the close environment configuration. It would be interesting if the authors try to take into account this climate change effect through new calculations.*

**RESPONSE 3.5**: The reviewer has a good point. In this study we are limited to a maximum depth of 1m, due to the patch continuity requirement. Knowing that the amplitude of soil moisture variations decreases with depth (Ravina, 1983), a drought that reaches deep soil layers will intensively dry out shallow layers. Theoretically, such an event can be detected with surface layer information alone. It should be noted, however, that drying out occurs with a time lag increasing with the depth of the

layer. We have verified in this analysis that even for the deepest model layers, the drought observed in a given year never overlaps with the following year, which would distort the index calculations. This assumption may be questioned under future climatic conditions, considering that the frequency and intensity of droughts in France will increase.

We added this to section 2.1:

"**In this study, we are limited to a maximum depth of 1 m due to the patch continuity requirement. One of the direct consequences of climate warming is the spread of deep soil desiccation under severe and recurrent drought conditions. This can reach soil layers deeper than 1 m depending on the close environment configuration. Knowing that the amplitude of soil moisture variations decreases with depth (Ravina, 1983), a drought that reaches deep soil layers will intensively dry out shallow layers. Theoretically, such an event can be detected with surface layer information alone. However, it should be noted that drying occurs with a time lag that increases with depth. In this analysis, we have verified that even for the deepest model layers, the drought observed in a given year never overlaps with the following year, which would distort the index calculations. This assumption may be questioned under future climatic conditions, considering that the frequency and intensity of droughts in France will increase.**"

**COMMENT 3.6**: *page 5 and line 127, what do the authors mean by "volumetric soil moisture"? Is it possible to explain how simulation can provide this physical property of the soil?*

**RESPONSE 3.6**: See Response 3.7.

**COMMENT 3.7**: *page 5 and lines 129-131, can the authors clarify better the "conversion" of the volumetric soil moisture to soil wetness indices (SWI) to justify considering a single definition of drought?*

**RESPONSE 3.7**: The Soil Wetness Index SWI consists of the soil moisture normalized between the field capacity $W_{fc}$ and the wilting point $W_{wilt}$, as expressed in the equation below. The latter two hydraulic parameters are derived from the texture using pedotransfer equations.

$$SWI = \frac{WG - W_{wilt}}{W_{fc} - W_{wilt}}$$

To justify our approach, we included the following figure Fig S1 in the supplementary material of the paper and added this sentence in Section 2.1:

"**While volumetric soil moisture is expressed in m3 m-3, SWI is unitless. SWI is derived by rescaling volumetric soil moisture between wilting point wwilt and field capacity wfc. as illustrated in Fig. S1 (see the Supplement).** "

[Figure]

Figure S1 – Unitless Soil Wetness Index (SWI) vs. volumetric soil moisture (WG) expressed in m3 m-3. Example of values simulated by the ISBA model for August 16, 2022 at 10:00 UTC for deciduous trees and 0.8-1.0 m soil layer: (c) WG, (d) SWI, and static (a) clay and (b) sand maps. For each grid cell, SWI results from the rescaling of WG between field capacity (Wfc) and wilting point (Wwilt) values derived from texture-dependent pedotransfer functions: SWI = (WG-Wwilt)/(Wfc-Wwilt).

**COMMENT 3.8**: *page 6 and lines 179-181, it appears that this study is mainly based on SWI outputs of the ISBA model. I'm not convicted that these calculations are the most reliable tools for studying soil moisture variations as mentioned.*

**RESPONSE 3.8**: Land Surface Models that simulate soil moisture variables are reliable tools for evaluating changes on a large scale. As explained in response to Comment 3.7, the conversion to SWI enables to solely focus on variations by removing the dependence on soil texture. Although the conversion is useful for unifying the information on a national scale, it does not affect the magnitude calculations. In fact, the calculations are based on thresholds defined by percentiles of the daily soil moisture, the distribution of which is unchanged by the linear transformation that is the conversion to SWI.

The following paragraph was added at the end of section 2.4:

"**It should be noted that although the conversion from volumetric soil moisture to SWI is useful for unifying the information on a national scale (Fig. S1), it does not affect the magnitude calculations. In fact, the calculations are based on thresholds defined by percentiles of daily soil moisture, the distribution of which is unchanged by the linear transformation that is the conversion to SWI.**"

and "the most reliable tools" was replaced by "**reliable tools**".

**COMMENT 3.9**: *page 6 and lines 189-190, I'm not sure that it is possible to assume that results based on in situ observations in the USA and Canada can be applicable to France especially under climate*

*change context. Many assumptions are considered in this study, which show the complexity to approach the soil water content and its variations without taking into account its hydromechanical properties at a given initial state.*

**RESPONSE 3.9**: We agree with the reviewer's comment. We are conscious that the hypothesis of a stable daily soil moisture distribution is questionable under a climate change context. We propose stating this in the discussion part dealing with uncertainties, section 4.4.3:

*'**However, there is an advantage to using daily instead of annual soil moisture data: the SWI distribution is expected to be more robust over a 19-year period due to the large number of observations. This is particularly important in the context of a changing climate. The hypothesis of a stable daily soil moisture distribution remains a source of uncertainty inherent to this work.**'*

**Additional references**

Burnol, A., Foumelis, M., Gourdier, S., Deparis, J., and Raucoules, D.: Monitoring of expansive clays over drought-rewetting cycles using satellite remote sensing, Atmosphere, 12, https://doi.org/10.3390/atmos12101262, 2021.

Clapp, R. B. and Hornberger, G. M.: Empirical equations for some soil hydraulic properties, Water Resour. Res., 14, 601–604, https://doi.org/10.1029/WR014i004p00601, 1978.

Crosetto, M., Solari, L., Balasis-Levinsen, J., Bateson, L., Casagli, N., Frei, M., Oyen, A., Moldestad, D. A., and Mróz, M.: Deformation monitoring at european scale: The copernicus ground motion service, in: International Archives of the Photogrammetry, Remote Sensing and Spatial Information Sciences - ISPRS Archives, 141–146, https://doi.org/10.5194/isprs-archives-XLIII-B3-2021-141-2021, 2021.

Decharme, B., Boone, A., Delire, C., and Noilhan, J.: Local evaluation of the Interaction between Soil Biosphere Atmosphere soil multilayer diffusion scheme using four pedotransfer functions, J. Geophys. Res. Atmospheres, 116, 1–29, https://doi.org/10.1029/2011JD016002, 2011.

Meisina, C., Zucca, F., Fossati, D., Ceriani, M., and Allievi, J.: Ground deformation monitoring by using the Permanent Scatterers Technique: The example of the Oltrepo Pavese (Lombardia, Italy), Eng. Geol., 88, 240–259, https://doi.org/10.1016/j.enggeo.2006.09.010, 2006.

Noilhan, J. and Lacarrère, P.: GCM Grid-Scale Evaporation from Mesoscale Modeling, J. Clim., 8, 206–223, https://doi.org/10.1175/1520-0442(1995)008<0206:GGSEFM>2.0.CO;2, 1995.

Ravina, I.: The influence of vegetation on moisture and volume changes, Géotechnique, 33, 151–157, https://doi.org/10.1680/geot.1983.33.2.151, 1983.

Tzampoglou, P., Loukidis, D., and Koulermou, N.: Seasonal Ground Movement Due to Swelling/Shrinkage of Nicosia Marl, Remote Sens., 14, 1440, https://doi.org/10.3390/rs14061440, 2022.